# Internet search analysis on the treatment of rheumatoid arthritis: What do people ask and read online?

**Satoshi Yamaguchi**[1,2]*, **Seiji Kimura**[2], **Shotaro Watanabe**[2], **Yukio Mikami**[2],
**Hirofumi Nakajima**[2], **Yukiko Yamaguchi**[2], **Takahisa Sasho**[2,3], **Seiji Ohtori**[2]

**1** Graduate School of Global and Transdisciplinary Studies, Chiba University, Chiba-shi, Chiba, Japan,
**2** Department of Orthopaedic Surgery, Graduate School of Medical and Pharmaceutical Sciences, Chiba University, Chiba-shi, Chiba, Japan, **3** Center for Preventive Medical Sciences, Chiba University, Chiba-shi, Chiba, Japan

* y-satoshi@faculty.chiba-u.jp

## Abstract

### Objectives

This study aimed to characterize the content of frequently asked questions about the treatment of rheumatoid arthritis (RA) on the internet in Japan and to evaluate the quality of websites related to the questions.

### Methods

We searched terms on the treatment of RA on Google and extracted frequently asked questions generated by the Google "people also ask" function. The website that answered each question was also obtained. We categorized the questions based on the content. The quality of the websites was evaluated using the brief DISCERN, Journal of American Medical Association benchmark criteria, and Clear Communication Index.

### Results

Our search yielded 83 questions and the corresponding websites. The most frequently asked questions were regarding the timeline of treatment (n = 17, 23%) and those on the timeline of the clinical course (n = 13, 16%). The median score of brief DISCERN was 11 points, with only 7 (8%) websites having sufficient quality. Websites having sufficient quality based on the Journal of American Medical Association benchmark criteria and Clear Communication Index were absent.

### Conclusions

The questions were most frequently related to the timeline of treatment and clinical course. Physicians should provide such information to patients with RA in the counseling and education materials.

**Data Availability Statement:** All relevant data are within the paper and its Supporting information files.

**Funding:** The author(s) received no specific funding for this work.

**Competing interests:** The authors have declared that no competing interests exist.

## Introduction

Patient education is a vital part of the management of rheumatoid arthritis (RA) and is recommended throughout the course of the disease [1]. To achieve patient-centered education, understanding patient needs regarding the treatment of RA is critical [1]. Meeting the information needs of patients and those who support the patients is positively associated with patient satisfaction and quality of life [2]. However, up to 80% of RA patients expressed needs for additional information [2]. Furthermore, the information needs of the patients and the public may not always align with those of physicians [1, 3, 4]. This discrepancy is particularly an issue because the treatment strategy and available pharmacological agents are changing dramatically in the past decade [3]. Therefore, a continued effort has been made to clarify the patient and public needs for RA treatment [5].

Various methods have been used to explore the information needs of patients with RA and people providing support to the patients, and they include questionnaire surveys, individual interviews, focus group discussions, and observation of the outpatient clinic [1, 3, 5–7]. The reported needs of patients vary widely across studies. One of the possible reasons for this variation is a small number of participants, which is an inherent limitation of individual interviews and focus group discussions [1, 6]; therefore, data from a larger number of people and a wider geographic area might provide information that would be more universally applicable in clinical practice.

Internet search results have been recently used to clarify common medical requirements [8]. This is based on the fact that people increasingly obtain health information from the internet. For example, 83% of internet users seek health information online in Japan [9]. Especially in chronic diseases such as RA, where a large part of the disease management occurs at home, the internet can be a vital resource for patients to obtain useful information [2, 6]. People also ask (PAA) is a function of the Google web search that provides a list of commonly asked questions related to the original search query. It also links to the website that attempts to answer each question. Google assesses the intent of a search query and sorts through billions of websites using a machine learning system and a natural language processing system that accurately detect search patterns [10, 11]. Therefore, using the Google PAA, we can retrieve the most commonly asked questions and associated websites concerning specific medical conditions. The PAA analysis has been used to clarify the public's questions on total joint arthroplasty, sinus surgery, and COVID-19 vaccines [8, 12, 13]. However, studies analyzing the frequently asked questions on the treatment of RA using PAA are lacking.

Well-designed online patient information can help patients understand the management of their disease and enhance patient-physician communication [14]. The quality of patient information on the internet, including clarity, transparency, and readability, can be assessed using several measurements [15, 16]. Studies have assessed the quality of English websites on the treatment of RA [17, 18]. However, the quality of Japanese websites has rarely been reported [19, 20]. Moreover, the quality of Japanese websites on RA treatment has not yet been evaluated.

The purposes of this study were to characterize the content of frequently asked questions on the internet about the treatment of RA and to evaluate the quality of websites related to the questions.

## Materials and methods

### Search strategy

This study was exempted from ethics review by the Research Ethics Committee of the Graduate School of Medicine, Chiba University because no patient data were used. All relevant data

are within the manuscript and its Supporting Information files. This study complies with the Declaration of Helsinki. We performed an internet search on April 21, 2022, in Chiba City, Japan (internet protocol address, 10.106.102.246) using a clean-installed Google Chrome browser (version 99.0.4844.51) to minimize the influence of search history on the results. The following six terms were searched: "rheumatoid arthritis" "treatment" ("関節リウマチ" "治療" in Japanese), "rheumatism"" treatment" ("リウマチ" "治療"), "rheumatoid arthritis" "pharmacotherapy" ("関節リウマチ" "薬物療法"), "rheumatism" "pharmacotherapy" ("リウマチ" "薬物療法"), "rheumatoid arthritis" "drugs" ("関節リウマチ" "薬"), and "rheumatism" "drugs" ("リウマチ" "薬"). In this study, we focused on the treatment rather than RA in general because it is the most commonly searched information among RA patients [2, 18]. The PAA questions for these six terms were extracted from Google searches using a free Chrome extension program (cited 2022 April 28, available from: https://seominion.com/). This program provides a non-personalized list of questions in PAA and the information on the website that answers the question. We used the top 100 questions per search term based on previous studies that used a similar methodology [21].

After the extraction of data, duplicate questions from individual searches were deleted. Subsequently, questions unrelated to RA were removed. Additionally, those that addressed other aspects of RA, such as pathology, epidemiology, symptoms, diagnosis/examination, and others, were removed. The remaining questions pertaining to the treatment of RA were analyzed.

## Classification of questions in PAA

As reported in the previous studies [8, 12, 13], the questions in PAA were first classified into three broad categories: fact, policy, and value, using the Rothwell classification of questions (Table 1) [22]. Fact questions were further sub-classified into six groups: general, mechanism, timeline of treatment, technical details, and cost. Policy questions were sub-classified into two groups: indication and risk or complication. Value questions were sub-classified into timeline of clinical course, prognosis, and evaluation. These sub-classifications were determined following the previous study; however, modifications were made to fit the topic of this study (Table 1) [8, 12, 13].

## Evaluation of answer source websites

The type of answer source websites was classified into six categories: academic, commercial, government, medical practice, single medical doctor, and social media (Table 1) [12]. The quality of the websites was evaluated using validated tools: the brief DISCERN, Journal of American Medical Association (JAMA) benchmark criteria, and Clear Communication Index (CCI). The brief DISCERN is a short version of the original DISCERN and comprises six questions regarding clarity, balance, and content of patient information [23]. Each question was rated between 1 and 5, and the total score ranged from 5 (worst) to 30 (best) points. Scores $\geq 16$ points were classified as having good content quality [23]. Information transparency of each website was assessed using the JAMA benchmark criteria. It used four criteria to evaluate websites: authorship, attribution, disclosure, and currency [15]. One point was given for each satisfied criterion, resulting in possible scores ranging from 0 to 4. Scores $\geq 3$ points were classified as having sufficient quality [15]. The CCI is a tool used to evaluate public communication products [16]. It comprises 20 items divided into seven themes: main message and call to action, language, information design, state of science, recommendation for action, numbers, and risk. Each item is scored as 0 or 1, and the percentage values between 0 and 100 points are calculated. A score $\geq 90$ indicated that the material was easy to read and conveyed the information well.

**Table 1. Rothwell classification of questions, question sub-classification, and answer source website categorization.**

| Rothwell classification | Description |
|---|---|
| Fact | Asks objective, factual information |
| Policy | Asks information on a specific course of action that should be taken to solve a problem |
| Value | Asks for evaluation of an idea, object, or event |
| **Question sub-classification** | **Description** |
| Fact | |
| General information | General questions not categorized below |
| Mechanism | Mechanism of treatment |
| Timeline of treatment | Questions regarding the length of time for treatment |
| Technical details | Treatment procedures on medications |
| Cost | Cost of treatment |
| Policy | |
| Indication | Indication of treatment |
| Risk or complication | Risks or complications during treatment |
| Value | |
| Timeline of clinical course | Questions regarding the length of time for clinical course |
| Prognosis | Questions regarding the consequence of treatment |
| Evaluation | Evaluation of treatment |
| **Answer source website categorization** | **Description** |
| Academic | Academic institutions, including universities, academic medical centers, and academic societies |
| Commercial | Commercial organizations, such as medical device and pharmaceutical companies |
| Government | Websites ending in ".gov" or maintained by government organizations |
| Medical practice | Local hospitals or orthopedic practices without an academic affiliation |
| Single medical doctor | Websites maintained by individual medical doctors without an institutional affiliation |
| Social media | Nonmedical organizations and websites intended for information sharing |

## Statistics

Descriptive statistics were used to express the results of question classification and answer source website evaluation. Categorical values were shown using numbers and percentages. Continuous values were expressed using the median and percentiles because they had non-normal distribution. The assessments of the website quality were compared among the answer source categories using the Kruskal–Wallis tests and post hoc Steel–Dwass tests. To assess the intra-rater reliability of the question sub-classification and website evaluations, a health professional (YY) assessed 20 randomly selected websites twice at an interval of four weeks. The kappa statistics and intra-class correlation coefficient were calculated as appropriate. To assess inter-rater reliability, an orthopaedic surgeon (SY) performed the assessment once, and the results were compared with those of the first assessment by the first rater. Statistical significance was set at $P < 0.05$.

## Results

Of the 600 frequently asked questions extracted from the six search queries, 172 questions were on RA. Of those, 89 were unrelated to the treatment of RA. The remaining 83 questions on RA treatment were analyzed (Fig 1).

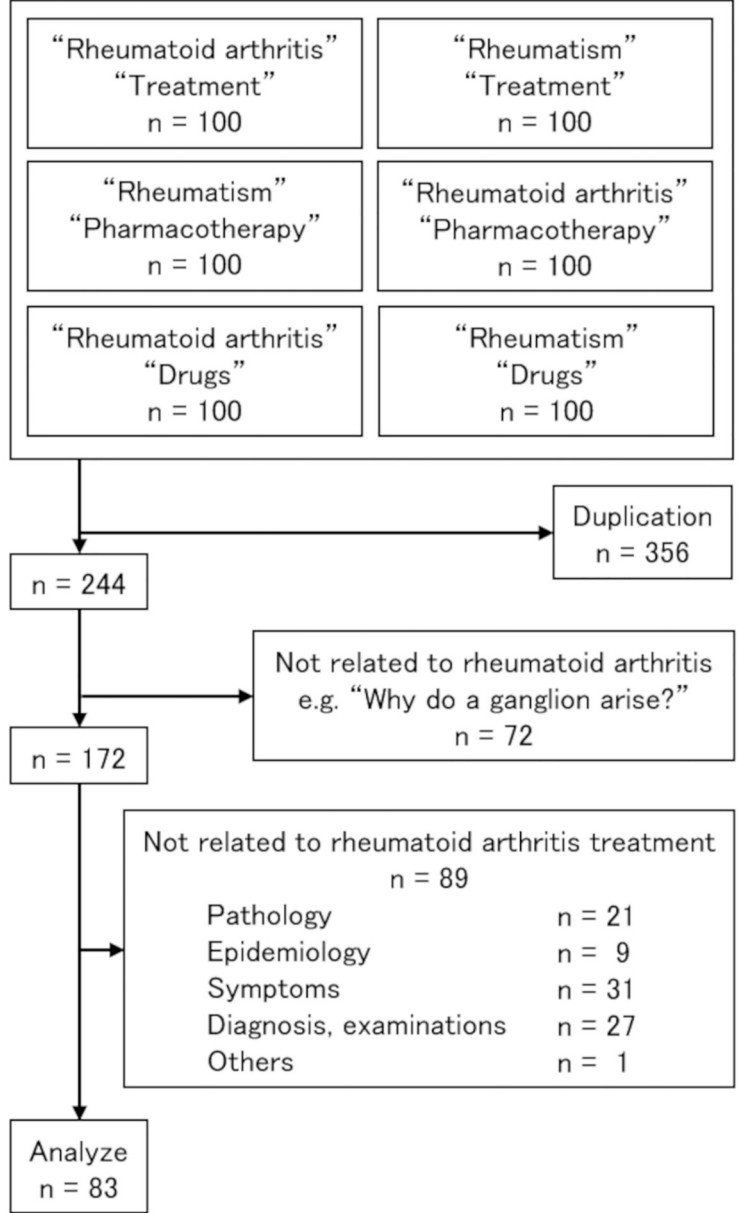

**Fig 1. Flow chart of website selection.** The top square indicates the initial website selection. The lower squares show the excluded websites and the numbers for analysis.

According to the Rothwell classification, 50 (60%) questions were categorized into fact questions, followed by value and policy (Table 2). In the sub-classification by question topic, the most frequent topic was the timeline of treatment, such as "How long should I take methotrexate?" (n = 17, 20%, Table 2 and S1 Table). Questions on the timeline of the clinical course (n = 13, 16%) and prognosis (n = 13, 16%) were also common. Questions were not classified into indication or evaluation. Examples of questions in each sub-classification are presented in Table 2 and S1 Table. When analyzing individual questions, the most common question was about the discontinuation of treatment, including "What happens if I don't treat my rheumatism" and "What if I stop drug treatment?" (n = 8).

**Table 2. Question classification (n = 83).**

| Rothwell classification | | n (%) | Example |
|---|---|---|---|
| | Question sub-classification | n (%) | |
| Fact | | 50 (60) | |
| | General information | 13 (16) | Are there any medications that work for rheumatism? How can rheumatism be treated? |
| | Mechanism | 13 (16) | Methotrexate / Folic acid / Why combined? Prednisone / Why morning? |
| | Timeline of treatment | 17 (20) | Until when do I continue rheumatoid medications? Rheumatism / Injection / Until when? |
| | Technical details | 4 (5) | Rheumatism / Injection / Where? |
| | Cost | 3 (4) | Adalimumab / How much? |
| Policy | | 7 (8) | |
| | Indication | 0 (0) | |
| | Risk or complication | 7 (8) | Methotrexate / Side effects / From when? |
| Value | | 26 (31) | |
| | Timeline of clinical course | 13 (16) | Tocilizumab / Effect / From when? Rheumatism / Joint destruction / From when? |
| | Prognosis | 13 (16) | Can rheumatoid arthritis be cured? Rheumatism / Leave untreated / What happens? |
| | Evaluation | 0 (0) | |

More than half of the answer source websites were categorized as medical practice (n = 46, 55%, Table 3). Only 17 (20%) were academic websites. Government and single medical doctor websites were absent. Overall, the quality of the answer source websites was poor. The median brief DISCERN score was 11, with only 7 (8%) websites having sufficient quality with $\geq 16$ points (Table 3). The median JAMA benchmark criteria score was 0 points, and the median CCI score was 40 points (Table 3). Websites classified as having sufficient quality based on these measurements were absent. Regarding the domain scores of the measurements, those on information transparency, such as the Brief DISCERN questions 1 and 2, all JAMA benchmark criteria questions, and CCI question 11, were low (S2 Table). Additionally, scores were low on questions related to the overall quality of life (brief DISCERN question 1) and main message and call to action (CCI questions 1–5). However, the scores of the other domains were also not sufficient.

The quality measurement scores were significantly different depending on the answer source categorization ($P < 0.001$ for the brief DISCERN, $P < 0.001$ for the JAMA benchmark

**Table 3. Website quality according to answer source category.**

| | Overall (n = 83) | Academic (n = 17) | Commercial (n = 8) | Medical practice (n = 46) | Social media (n = 12) | P |
|---|---|---|---|---|---|---|
| Brief DISCERN | 11 (8, 13) | 12 (10, 13) | 14 (11, 14) | 9 (8, 12)[a] | 17 (9, 19) | < 0.001 |
| JAMA | 0 (0, 1) | 0 (0, 1) | 1 (1, 1) | 0 (0, 1)[b] | 1 (1, 1) | < 0.001 |
| CCI | 40 (40, 47) | 40 (30, 52) | 40 (27, 40)[c] | 42 (40, 47) | 49 (43, 50) | 0.01 |

Values indicate the median (25th, 75th percentiles). Significant differences between

[a]the social media websites,

[b]academic, commercial, and social media websites, and

[c]social media websites in the post hoc Steel-Dwass tests ($P < 0.05$).

JAMA, Journal of the American Medical Association benchmark criteria; CCI, Clear Communication Index.

**Table 4. Intra- and interrater reliability of question and website assessments.**

|  | Question sub-classification[a] | Website category[a] | Brief DISCERN[b] | JAMA[b] | CCI[b] |
| --- | --- | --- | --- | --- | --- |
| Intrarater | 0.94 (0.89, 0.99) | 1.00 | 0.94 (0.85, 0.97) | 0.89 (0.76, 0.96) | 0.87 (0.71, 0.95) |
| Interrater | 0.75 (0.63, 0.84) | 0.86 (0.67, 1.00) | 0.92 (0.81, 0.97) | 0.94 (0.87, 0.98) | 0.87 (0.71, 0.95) |

Values indicate

[a]kappa statistics and

[b]intraclass correlation coefficient (95% confidence intervals).

JAMA, Journal of the American Medical Association benchmark criteria; CCI, Clear Communication Index.

criteria, and $P = 0.01$ for the CCI, Table 3). The social media websites scored the highest in all three measurements (Table 3). The brief DISCERN score was the lowest in the medical practice websites, and the CCI was lowest in the commercial websites (Table 3).

The intra- and inter-rater reliability of question and website assessments were sufficient, with the kappa statistics and intra-class correlation coefficient values ranging between 0.75 and 1.00 (Table 4).

## Discussion

This study showed that Japanese people frequently searched for the timelines of treatment and clinical course of RA on the internet. People also asked about the course of the disease if they stopped treatment. More than half of the answer source websites were medical practice websites, whose quality of information was generally low. The results of our study will help better address the public's information needs regarding the treatment of RA.

In this study, majority of the questions in PAA were regarding the timeline of treatment and clinical course. Similar to our study, questions regarding the timeline of recovery following surgical procedures were commonly asked online [12, 13]. Our results also agree with a previous study that reported that patients with depression commonly needed online information concerning treatment timelines [24]. However, previous studies on patients with RA using a focus group discussion and questionnaire survey did not identify the timeline of treatment as an information need [1, 7]. These studies found other unmet information needs, such as medication side effects and diet [5, 7]. Therefore, analysis using PAA, which is a sum of the vast number of search results, may reveal the needs of patients and the public that are not assessed by individual interviews and small-group discussions. Furthermore, we searched Japanese words that were exclusively used domestically. Therefore, the results of this study highlighted the information needs of Japanese people, which may differ from those of people in other countries. Therefore, physicians and medical providers need to incorporate information on the treatment timelines into counseling and education resources for RA patients.

We showed that people frequently asked about the consequence of absence of treatment or discontinuation of treatment. Our results agreed with those of previous studies that discontinuation of treatment was the preferred topic to share for RA patients [1, 5]. Although promoting adherence to treatment is challenging, poor treatment adherence is linked to increased disease activity and may lead to complications [25]. Therefore, healthcare providers should inform RA patients regarding this topic.

In this study, academic websites accounted for only 20% of the answer source. Furthermore, government websites were absent. Although the type of website that answered the PAA may differ depending on the search query, the result of our study is in contrast with the studies of English websites, in which about 40% of the answer source websites on common diseases were

government and academic websites [8, 12]. A lack of easily accessible and reliable online patient resources provided by government bodies has been an issue in Japan [26].

The quality of the answer source websites was poor, with only 8% of them meeting the criteria for sufficient quality based on the brief DISCERN. Websites satisfying the quality criteria for the JAMA benchmark criteria and CCI were absent. Our results were consistent with the results of previous studies on Japanese websites, in which online patient resources on musculoskeletal diseases had low quality [19, 20]. Furthermore, the quality of internet information on osteoarthritis in Japan was lower than those of other countries [27]. However, insufficient quality of online patient information has been reported regardless of the language, region, and medical field [28]. For example, about 80% of English websites on the education of RA patients did not address the latest date of updating of the content [17]. Good-quality healthcare websites can fulfill patient information needs, improve treatment adherence, and facilitate shared decision-making [28]. Several academic societies, such as the American Academy of Orthopaedic Surgeons, have been trying to improve the quality of their websites through repeated evaluations [14]. Furthermore, the assessment tools in this study can be used to improve the quality of patient information [29, 30]. Therefore, healthcare providers involved in RA treatment, need to recognize the importance of quality when creating and updating online patient resources.

In this study, the overall quality of the social media websites appeared to be higher than that of the commercial and medical practice websites. The association between the website category and quality is conflicting. A study on the websites on COVID-19 reported that the brief DISCERN score of the social media websites was lower than that of the government websites [8]. Another study reported that the quality of academic websites was the lowest among the internet-based information on osteoporosis [31]. A possible explanation of the discrepancy between the studies is the differences in the classification method, topic, and region of the studies. However, even the social media websites in this study had lower quality than previously reported websites [8]. Furthermore, the evaluation tools used in this study did not assess the correctness of the information. Therefore, even websites rated as high quality may contain incorrect information. All medical providers, regardless of the category, should provide high-quality websites on the treatment of RA to facilitate patient-physician communication.

This study has several limitations. First, it is unknown what period of internet information was used to generate the PAA search results presented by Google. Furthermore, the results may change if more query data were collected over time or if the data sorting algorithm changed in the future. The search results were also different depending on the language used and the region where the search was performed. Although this study focused on the information needs of Japanese people, further studies are necessary to clarify the patient needs in other regions. Second, although we used non-personalized data on the questions and related websites, they may differ from person to person because individual search history could affect the results. Third, although the questions in PAA were ideally from potential RA patients and related people, we do not know who asked the questions. Thus, we can consider that our results represent what the public wants to know about RA treatment. Fourth, the search engine market share of Google is about 77% in Japan [32]. Therefore, the search data using other search engines were not included in our results. Fifth, Internet search results may vary depending on the search terms. We searched for general terms such as "treatment," similar to the previous studies [8, 12, 13, 18]. However, searching for more specific words, such as "biologics," may reveal more specific patient needs at the cost of limited generalizability. Finally, this study was conducted during the Coronavirus disease 2019 pandemic, when health information-seeking behavior on the internet might differ from that of the pre-pandemic period [33]. Therefore, a comparison between the results of this study and previous studies should be made with caution.

In conclusion, most of the questions asked by patients in PAA were regarding the timeline of treatment. Physicians should share information regarding the treatment and clinical timeline with RA patients. Furthermore, the quality of the answer source websites was insufficient. Especially, descriptions of information transparency were lacking. Medical providers involved in managing RA need to provide better-quality online patient resources to facilitate patient-physician communication.

## Supporting information

**S1 Table. Question classification (in Japanese, n = 83).**
(DOCX)

**S2 Table. Domain scores of the website quality measurements (n = 83).**
(DOCX)

**S3 Table. Question classification, answer source category, and quality measurements of each question (n = 83).**
(XLSX)

## Author Contributions

**Conceptualization:** Satoshi Yamaguchi.

**Data curation:** Satoshi Yamaguchi, Shotaro Watanabe, Yukio Mikami, Hirofumi Nakajima, Yukiko Yamaguchi.

**Formal analysis:** Satoshi Yamaguchi, Shotaro Watanabe.

**Investigation:** Seiji Kimura, Yukio Mikami, Hirofumi Nakajima.

**Methodology:** Satoshi Yamaguchi, Seiji Kimura.

**Supervision:** Takahisa Sasho, Seiji Ohtori.

**Validation:** Satoshi Yamaguchi, Yukiko Yamaguchi.

**Writing – original draft:** Satoshi Yamaguchi, Seiji Kimura.

**Writing – review & editing:** Satoshi Yamaguchi, Takahisa Sasho, Seiji Ohtori.

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
