## [Decision Letter · Decision Letter 0]

2 Mar 2023

PONE-D-23-01919Internet search analysis on the treatment of rheumatoid arthritis: What do people ask and read online?PLOS ONE

Dear Dr. Yamaguchi,

Thank you for submitting your manuscript to PLOS ONE. After careful consideration, we feel that it has merit but does not fully meet PLOS ONE’s publication criteria as it currently stands. Therefore, we invite you to submit a revised version of the manuscript that addresses the points raised during the review process.

The paper was well written following a standard format. Please address the concerns raised by reviewers prior to the final decision.

We look forward to receiving your revised manuscript.

Kind regards,

Yoshito Nishimura, MD, PhD, MPH

Academic Editor

PLOS ONE

Journal Requirements:

2. In your Methods section, please include additional information about your dataset and ensure that you have included a statement specifying whether the collection and analysis method complied with the terms and conditions for the source of the data

Additional Editor Comments (if provided):

Thank you for your submission. Please revise the manuscript per reviewers' comments.

Reviewers' comments:

Reviewer's Responses to Questions

**Comments to the Author**

1. Is the manuscript technically sound, and do the data support the conclusions?

Reviewer #1: Yes

Reviewer #2: Yes

2. Has the statistical analysis been performed appropriately and rigorously? 

Reviewer #1: Yes

Reviewer #2: Yes

3. Have the authors made all data underlying the findings in their manuscript fully available?

Reviewer #1: Yes

Reviewer #2: Yes

4. Is the manuscript presented in an intelligible fashion and written in standard English?

Reviewer #1: Yes

Reviewer #2: Yes

5. Review Comments to the Author

Reviewer #1: This is an interesting study summarizing the reliability of information on the Internet regarding the treatment of rheumatoid arthritis. Since the Japanese language is used in Japan, there are relatively fewer sources of information than in other languages, and patients with limited media literacy often believe incorrect information. The data can objectively demonstrate this situation, however, I suggest some revisions for publication.

As the authors point out, the fact that only Japanese is used is a limitation, but on the other hand, Japanese is a unique language in the world, used almost exclusively in Japan and as a native language by most Japanese people, thus the results of this study may represent a one-to-one correspondence with domestic trends in Japan. From this point of view, it would be possible to discuss the differences between the results in Japan and other countries.

The study was conducted in the midst of COVID-19 pandemic. It is reported that it was difficult to receive out-patient treatment for chronic diseases during COVID pandemic, and it is possible that patients sought information sources on the Internet. This point should be discussed.

As for the searching term, it would be appropriate to indicate the words actually inputted in kanji characters instead of romaji.

Figure Legend should be added as it seems to be missing.

Please state the official name of the IRB which not requiring ethics review.

Please tell us the version of Google Chrome you used.

Google may change the search results depending on where you accessed the Internet, so please share information such as place names and IP addresses rather than simply stating "Japan" if possible

Reviewer #2: I was glad to review this article which was really interesting.

I suggested a few point to revise as below.

Regarding the terms such as ''Single surgeon personal'', ''Websites maintained by individual surgeons without an institutional affiliation'' especially in Table 3, I know some area orthopedic surgeon is treating rheumatoid arthritis. However, those patients are usually treated by rheumatologist(or physician) in general all over the world. Therefore, this term may need to be changed to other phrase. For instance, single physician or surgeon personal or single medical doctor, which include physicians and surgeon.

Regarding in Table 2, there was a word ''Humira''. It should be changed to generic name, Adalimumab''.

Lastly, for all the tables, probably it can be sorted clearly so subscribers understand better.

6. PLOS authors have the option to publish the peer review history of their article (what does this mean?). If published, this will include your full peer review and any attached files.

Reviewer #1: **Yes: **Yuki Otsuka

Reviewer #2: No

---

## [Author Response · Author response to Decision Letter 0]

3 Mar 2023

Responses are in the attached file

---

## [Decision Letter · Decision Letter 1]

5 Apr 2023

PONE-D-23-01919R1Internet search analysis on the treatment of rheumatoid arthritis: What do people ask and read online?PLOS ONE

Dear Dr. Yamaguchi,

Thank you for submitting your manuscript to PLOS ONE. After careful consideration, we feel that it has merit but does not fully meet PLOS ONE’s publication criteria as it currently stands. Therefore, we invite you to submit a revised version of the manuscript that addresses the points raised during the review process.

We look forward to receiving your revised manuscript.

Kind regards,

Yoshito Nishimura, MD, PhD, MPH

Academic Editor

PLOS ONE

Journal Requirements:

Additional Editor Comments:

Dear Authors, thank you for submitting your revision. While the manuscript has been improved according to the comments from 2 reviewers, additional comments were brought up to further secure the quality of the manuscript. Please refer to the comments by reviewer 3. I look forward to your revision.

Reviewers' comments:

Reviewer's Responses to Questions

**Comments to the Author**

1. If the authors have adequately addressed your comments raised in a previous round of review and you feel that this manuscript is now acceptable for publication, you may indicate that here to bypass the “Comments to the Author” section, enter your conflict of interest statement in the “Confidential to Editor” section, and submit your "Accept" recommendation.

Reviewer #1: All comments have been addressed

Reviewer #2: All comments have been addressed

Reviewer #3: (No Response)

2. Is the manuscript technically sound, and do the data support the conclusions?

Reviewer #1: Yes

Reviewer #2: Yes

Reviewer #3: Partly

3. Has the statistical analysis been performed appropriately and rigorously? 

Reviewer #1: Yes

Reviewer #2: Yes

Reviewer #3: Yes

4. Have the authors made all data underlying the findings in their manuscript fully available?

Reviewer #1: Yes

Reviewer #2: Yes

Reviewer #3: Yes

5. Is the manuscript presented in an intelligible fashion and written in standard English?

Reviewer #1: No

Reviewer #2: Yes

Reviewer #3: Yes

6. Review Comments to the Author

Reviewer #1: The manuscript is well addressed, however, the manuscript should be proofread by a native English speaker.

Reviewer #2: I was my pleasure to review this article. All minor reviews were appropriately addressed. I agree with publishing this.

Reviewer #3: I enjoyed reading your article. The following are my comments after reviewing it.

- This PAA system reflects the tendency of questions of what time period? Today? For the last 1 week? For the last one month?

Does it reflect the general tendency of questions correctly or is there a possibility that the results may be affected by one temporary "phase" of search which is affected by a recent popular TV program for example? I think if the data sampling period is too short, there is the risk of misinterpreting a random rise in the search number as a general trend.

- I assume google search system tries to direct viewers to a website that gives quick and easy answer to each question. From that perspective, the websites that were linked to PAA questions do not necessarily have to be detailed, high quality or thorough regarding the general topic such as "treatment in general". You concluded that the quality of websites on RA treatment was insufficient. However, from this study, this big conclusion cannot be obtained because you did not collect the websites that explains "RA treatment in general." You assessed websites that "answered each commonly asked question" and assessed their quality. If you want to argue that "websites on RA treatment has poor quality", you would have to directly go to the websites that pop up with the general search "rheumatoid arthritis/treatment" etc, not the website that were linked with PAA. Once again, the websites that you picked up by PAA are the ones that likely give people "easy answers." for each small question.

- I think there is weakness of this search method. This study is based on the premise that people start searching with specific 6 combinations of terms. However, this is not always true in the real world. For example, if people searched with more specific terms such as "rheumatoid arthritis, biologics", worrying about the high cost of biologics, that tendency would not be reflected on the result of this study. Others may search with "rheumatoid arthritis, side effects" or "rheumatoid arthritis, methotrexate" "rheumatoid arthritis, infectious risk". Many patients often have specific concerns and they tend to search with more specific terms depending on their concerns.

Therefore, this study dose shed light to one aspect of people's concerns affected by random bias from the search engine but does not reflect the real-life frequency of the general population. Actual interview style studies from a smaller RA cohort would have less bias, reflecting the true tendency of question.

- Are there any websites that were rated as "good" in Japan in general? Even academic websites that were linked failed to meet "good quality" criteria. Can this be a problem of the outcome measures that were used? Or can this be a misguidance by google to a low quality website? Or does this reflect poor quality of Japanese online resources in general?

If you think there might be a problem with the quality measures, I would want to hear more about why and in what way. (More details in the next section.) If you think it could be a google misguidance problem despite the existence of high quality websites in Japan, you might want to mention examples of websites with "good quality" in Japan. If you think this is because of the general paucity of such websites in Japan (although I do not believe this study can directly support this conclusion), you might want to elaborate on that.

- Regarding the validity of the quality measures, it concerns me that social media is rated as the highest quality compared with academic or medical practice websites because it is contraty to my intuition. I would imagine academic entities would provide more accurate, less biased and updated information than social media, which has no guarantee regarding bias or background knowledge of the authors. You did mention other studies using the same quality measures in the discussion. Do you think these measures appropriately evaluated the quality of RA websites? Do you think that academic website has information with lower quality than social media? If you think there is some problem with these quality measures, you should comment on that in the discussion because I think these questions are crucial to this paper. The reliability of outcome depends on the validity of these quality measures.

Overall, I liked this novel attempt in filling the knowledge gap between patients and physicians. I do think your messages here at least partially hold truth from personal clinical experience. However, I also think a few questions regarding logics and the design that were mentioned above should be addressed.

7. PLOS authors have the option to publish the peer review history of their article (what does this mean?). If published, this will include your full peer review and any attached files.

Reviewer #1: **Yes: **Yuki Otsuka

Reviewer #2: No

Reviewer #3: **Yes: **Shuhei Hattori

---

## [Author Response · Author response to Decision Letter 1]

9 Apr 2023

Responses were included in ”Response to Reviewers” file

---

## [Decision Letter · Decision Letter 2]

25 Apr 2023

PONE-D-23-01919R2Internet search analysis on the treatment of rheumatoid arthritis: What do people ask and read online?PLOS ONE

Dear Dr. Yamaguchi,

Thank you for submitting your manuscript to PLOS ONE. After careful consideration, we feel that it has merit but does not fully meet PLOS ONE’s publication criteria as it currently stands. Therefore, we invite you to submit a revised version of the manuscript that addresses the points raised during the review process.

We look forward to receiving your revised manuscript.

Kind regards,

Yoshito Nishimura, MD, PhD, MPH

Academic Editor

PLOS ONE

Journal Requirements:

**Additional Editor Comments:**

Thank you very much for all the time and efforts revising the manuscript. One reviewer pointed out a few issues that needed to be addressed prior to acceptance. Please review the comments and proceed with your edits.

Reviewers' comments:

Reviewer's Responses to Questions

**Comments to the Author**

1. If the authors have adequately addressed your comments raised in a previous round of review and you feel that this manuscript is now acceptable for publication, you may indicate that here to bypass the “Comments to the Author” section, enter your conflict of interest statement in the “Confidential to Editor” section, and submit your "Accept" recommendation.

Reviewer #1: All comments have been addressed

Reviewer #3: (No Response)

2. Is the manuscript technically sound, and do the data support the conclusions?

Reviewer #1: (No Response)

Reviewer #3: Yes

3. Has the statistical analysis been performed appropriately and rigorously? 

Reviewer #1: Yes

Reviewer #3: Yes

4. Have the authors made all data underlying the findings in their manuscript fully available?

Reviewer #1: Yes

Reviewer #3: Yes

5. Is the manuscript presented in an intelligible fashion and written in standard English?

Reviewer #1: Yes

Reviewer #3: Yes

6. Review Comments to the Author

Reviewer #1: The manuscript has been fully revised. All of my points have been addressed appropriately and faithfully. I have no additional comments. I believe it is worthy enough to be published.

Reviewer #3: Thank you for revising the manuscript. As I mentioned in the previous comments, overall, I had 2 major questions to be addressed. 1, uncertainty of google search system regarding sampling bias. 2, whether outcome measures for quality assessment of websites were appropriate or not.

You addressed both of these questions better in the discussion of the latest manuscript. Given that you admitted the limitations of the current methods, the conclusion appears less biased.

I have one more request regarding the second conclusion of the study. I would like to hear a little more details regarding the low quality of websites. I would like to see what elements are lacking. You state that websites should have better quality. You must have assessed the websites based on 3 scoring systems and you should know which part websites scored less than ideal. It would be better to show what improvements are to be made than to just say they have low quality.

Other than that, I have no additional comments to make.

7. PLOS authors have the option to publish the peer review history of their article (what does this mean?). If published, this will include your full peer review and any attached files.

Reviewer #1: **Yes: **Yuki Otsuka

Reviewer #3: **Yes: **Shuhei Hattori

---

## [Author Response · Author response to Decision Letter 2]

29 Apr 2023

Responses were attached to the submitted manuscript

---

## [Editor Report · Decision Letter 3]

4 May 2023

Internet search analysis on the treatment of rheumatoid arthritis: What do people ask and read online?

PONE-D-23-01919R3

Dear Dr. Yamaguchi,

We’re pleased to inform you that your manuscript has been judged scientifically suitable for publication and will be formally accepted for publication once it meets all outstanding technical requirements.

Kind regards,

Yoshito Nishimura, MD, PhD, MPH

Academic Editor

PLOS ONE

Additional Editor Comments (optional):

Thank you for all the hard work to submit your revised manuscript. The quality of manuscript has been substantially improved after the rounds of revision.
---

## [Editor Report · Acceptance letter]

16 May 2023

PONE-D-23-01919R3 

Internet search analysis on the treatment of rheumatoid arthritis: What do people ask and read online? 

Dear Dr. Yamaguchi:

I'm pleased to inform you that your manuscript has been deemed suitable for publication in PLOS ONE. Congratulations! Your manuscript is now with our production department. 

Kind regards, 

on behalf of

Dr. Yoshito Nishimura 

Academic Editor

PLOS ONE